# Explanatory models of stillbirth among bereaved parents in Afghanistan: Implications for stillbirth prevention

Aliki Christou[1,2]*, Camille Raynes-Greenow[2], Adela Mubasher[3], Sayed Murtaza Sadat Hofiani[4], Mohammad Hafiz Rasooly[5], Mohammad Khakerah Rashidi[6], Neeloy Ashraful Alam[2]

1 Department of Public Health, Institute of Tropical Medicine, Antwerp, Belgium, 2 Sydney School of Public Health, Faculty of Medicine and Health, The University of Sydney, Sydney, Australia, 3 World Health Organisation, Kabul, Afghanistan, 4 French Medical Institute for Mothers and Children (FMIC), Kabul, Afghanistan, 5 Ministry of Public Health, Kabul, Afghanistan, 6 Aga Khan Foundation, Kabul, Afghanistan

* achristou@itg.be

**Data Availability Statement:** The data cannot be made publicly available as this may compromise the respondents' privacy. The data may be

## Abstract

Local perceptions and understanding of the causes of ill health and death can influence health-seeking behaviour and practices in pregnancy. We aimed to understand individual explanatory models for stillbirth in Afghanistan to inform future stillbirth prevention. This was an exploratory qualitative study of 42 semi-structured interviews with women and men whose child was stillborn, community elders, and healthcare providers in Kabul province, Afghanistan between October-November 2017. We used thematic data analysis framing the findings around Kleinman's explanatory framework. Perceived causes of stillbirth were broadly classified into four categories—biomedical, spiritual and supernatural, extrinsic factors, and mental wellbeing. Most respondents attributed stillbirths to multiple categories, and many believed that stillbirths could be prevented. Prevention practices in pregnancy aligned with perceived causes and included engaging self-care, religious rituals, superstitious practices and imposing social restrictions. Symptoms preceding the stillbirth included both physical and non-physical symptoms or no symptoms at all. The impacts of stillbirth concerned psychological effects and grief, the physical effect on women's health, and social implications for women and how their communities perceive them. Our findings show that local explanations for stillbirth vary and need to be taken into consideration when developing health education messages for stillbirth prevention. The overarching belief that stillbirth was preventable is encouraging and offers opportunities for health education. Such messages should emphasise the importance of care-seeking for problems and should be delivered at all levels in the community. Community engagement will be important to dispel misinformation around pregnancy loss and reduce social stigma.

## Introduction

Evidenced-based interventions and approaches to reduce stillbirths are known and include access to quality antenatal and intrapartum care from a skilled health provider to identify high

requested by sending an email the University of Sydney ethics committee human.ethics@sydney.edu.au.

**Funding:** The data collection for this study was partly funded through the Sydney School of Public Health and a Grants-in-Aid fund from the University of Sydney. At the time this work was undertaken, AC was supported by an Australian Postgraduate Award (PhD Scholarship) received through the Australian Commonwealth Government and CRG was receiving salary as part of an Australian government National Health and Medical Research Council Career Development Fellowship (1087062). During the preparation of the manuscript, AC was receiving salary as part of a post-doctoral fellowship through the Flanders Research Foundation (FWO), Belgium (number 1294322N). The specific roles of these authors are articulated in the 'author contributions' section. The funders had no role in study design, data collection and analysis, decision to publish, or preparation of the manuscript.

**Competing interests:** The authors have declared that no competing interest exist.

risk women and treat complications in pregnancy [1, 2]. However, these interventions rely on active care-seeking, recognition of danger signs and awareness of the importance of accessing healthcare during pregnancy and childbirth—in addition to the availability of quality care. In Afghanistan and in many other low- and middle-income countries (LMIC), many women do not receive adequate care during pregnancy or childbirth, and a large proportion continue to birth at home, often without a skilled provider [3, 4]. This lack of, or delayed access to care contributes substantially to the 2 million stillbirths that occur globally every year, yet we have little understanding of how to address this challenge [5].

Understanding community perceptions of stillbirth and its causes can provide insight into the barriers or challenges related to (pregnancy) practices and delays in care-seeking and can inform behaviour change interventions for prevention to improve perinatal outcomes. Such beliefs and perceptions have implications for health promotion and public health messages. Social and cultural beliefs vary across settings and so require approaches that are relevant to the local context. A better understanding is needed of community views around stillbirth and pregnancy loss to identify behaviours that may be harmful or protective to develop and target health promotion messages at the individual and community level. What is also critical to understand are the psychosocial impacts of stillbirth and potential social consequences for women that increase their vulnerability to family violence or stigma, as these need to be taken into consideration when developing interventions for stillbirth prevention [6].

Anthropological and sociological studies show that women are aware of their vulnerability during pregnancy and engage in certain behaviours to protect themselves and their baby [7]. Underscoring these behaviours are beliefs about the causes of pregnancy complications and adverse outcomes [8]. Such explanatory models are useful to understand and describe these different views that exist across patient and medical providers' conceptions of the causes and experiences of disease [9, 10]. From the patient's perspective, explanatory models of health and disease are influenced by the broader social, and cultural context, psychological factors and from prior experiences. Whereas healthcare providers hold largely biomedical views that emphasise the biological and physical aspects of disease aetiology. The explanatory framework by Kleinman [9] proposes that when there are differing models of illness between patient and health provider, this affects adherence with medical advice and treatment outcomes. Perceptions of disease causation can also influence an individual or community's responsiveness to health promotion messages. Thus, explanatory models can be useful for informing programs or interventions aimed at improving the uptake of treatments or health promoting behaviours. Eliciting these explanatory models can generate knowledge on the social meanings attached to an illness or outcome and how individuals perceive, interpret and respond, and their expectations of what will happen. They have been used to understand mental illness, hypertension, diabetes and cancer particularly in settings and among population groups where western biomedical paradigms of illness are less predominant [11–13]. Stillbirth has not been studied using explanatory models.

In Afghanistan, stillbirth rates have remained high with estimated stillbirth rate in 2020 of 28 per 1000 births [5]. Our previous research has highlighted some of the contributing factors to stillbirth in Afghanistan including low levels of access to antenatal care, delays in, or not seeking care as well as concerns with quality of care [14]. Stillbirth prevention has not been addressed or prioritised at the national level and urgently needs attention. Here we seek to understand some of the perceptions and beliefs about the causes and prevention of stillbirth that might contribute to delays in seeking care or explain care practices in pregnancy that may be harmful and can inform future health promotion efforts at the community level.

The aim of this study was to explore the local perceptions of causes of stillbirth and the behaviours women engage in to avoid fetal harm or pregnancy loss. These explanatory models

would allow us to understand driving factors behind care-seeking and pregnancy practices to identify points for intervention. Furthermore, we also explore the psycho-social consequences and coping strategies used by bereaved parents as these can also guide the adaptation and targeting of health programs after a pregnancy loss.

## Methods

### Study design

We used an exploratory qualitative study design employing semi-structured, in-depth interviews to elicit the experiences and perceptions of stillbirth from parents and healthcare providers. Details of the study methodology have been previously published and briefly summarised below [15]. The data for this analysis are from a qualitative study that have not been presented in previous publications. The current study focuses primarily on respondents' views on the perceived causes of stillbirth and the preventative practices and behaviours women engage in to protect their pregnancy and prevent adverse outcomes. In addition, we explore the psycho-social consequences of stillbirth and coping strategies of bereaved parents.

### Study setting

The study was undertaken in urban and rural districts of Kabul province, Afghanistan in 2017. Study sites included three high-volume referral maternity hospitals in Kabul city, and two lower-level health facilities and surrounding communities in two rural districts ~25–30 kilometres west and north of Kabul.

### Participants and recruitment

Study participants comprised of women (n = 21) and men (n = 9) who had a recent stillborn baby, female community elders (n = 3), community health workers (CHW) (n = 5) and midwives (n = 4). Purposive sampling was used to recruit participants either through health facilities or from the community-level through CHWs and contacts of the local interviewers. We used several methods for recruitment of mothers who gave birth to a stillborn baby–firstly, we identified women either facility medical records and those with phone numbers available were contacted by telephone to invite them to participate; secondly, we received notification of stillbirths that occurred directly from healthcare providers at the health facilities included in our study; and thirdly, through the interviewers' community networks. Once mothers were identified we recruited fathers through those women and also through the networks of our interviewers'. We attempted to include women who gave birth at home as well as in the health facility but could only recruit five women (of 21 in total) who gave birth at home. In rural districts, CHWs assisted with identifying participants including female elders, however, we were only able to recruit three female elders in the time frame available for the data collection.

### Data collection

Interviews were conducted by three experienced Afghan qualitative interviewers (two female and one male). Interviewers participated in three days training on the research study, the interview guides and qualitative methods and interview techniques. We developed semi-structured in-depth interview guides for each participant type which explored their perceptions, understandings and experiences and practices related to stillbirth. Interview guides with women and men were pretested and modified where required prior to commencing data collection. After obtaining informed consent, interviews were conducted in the local language preferred by participants and audio recorded where permission was obtained. Interviews were held in private

locations chosen by participants and ranged in length from 30–60 minutes. Following each interview, interviewers completed a debrief form to document non-verbal observations, the interview environment, any new topics that arose, and challenges faced. This was followed by debriefing discussions with the study team members to discuss the data and findings.

## Data analysis

We conducted a thematic analysis of the data [16]. Analysis began with multiple readings of all transcripts followed by the generation of a code list. The coding framework was initially based on the interview guide topics and commenced with a narrative account of the respondent's experience with pregnancy loss and notions of what they believed caused stillbirth, the impacts or consequences on women and families following a stillbirth, and preventative practices engaged in to ensure a healthy pregnancy. Codes were then categorised and grouped into meaningful themes.

We applied the key domains of Kleinman's illness explanatory model [9] to thematically group codes into broader categories within this framework. Kleinman's framework comprises of several aspects of disease that make up an explanatory model including: definition of the illness, aetiology, symptoms, course of sickness, consequences, and treatment. We focused on the latter four categories to analyse our data and adapted Kleinman's domains for stillbirth creating five categories–i) perceived causes, ii) symptoms, iii) prevention practices, iv) consequences and impacts and v) treatment seeking and coping mechanisms. Prevention practices is a new category we have added that we feel provides important explanatory information. Perceived causes refer to beliefs about what respondents thought could result cause a stillbirth or what they have heard (and also what they felt might have contributed to their stillbirth); symptoms were signs preceding the stillbirth; prevention practices were those behaviours that women engage in to ensure a health pregnancy; consequences refers to reported social, psychological or physical impact a stillbirth had on women personally that could be physical or emotional; treatment seeking refers to any healthcare seeking *after* the stillbirth parents engaged in to avoid future pregnancy loss, while coping mechanisms refer to how parents responded and expressed their grief. Thematic categories were revised and refined after discussion among team members. We did not separate the analysis according to participant type as the data was overlapping and often participants would speak of what they know that others would do or believe (e.g., fathers might speak of what they thought their wife believed). It was not possible to separate individuals' perceptions from their experiences in the transcripts as responses reflected their perceptions about causes but sometimes also what they believed resulted in their pregnancy loss. We used N-Vivo 11 software to organise data into codes and themes for analysis.

## Ethical considerations

Ethical approval was provided by the institutional review board of the Afghanistan National Public Health Institute, Afghanistan (no. 43880) and the ethical review committee of the University of Sydney (no. 2017/566). Written permission was provided from participating hospitals and all participants gave written or verbal informed consent.

## Results

Participants' explanatory models for stillbirth are summarised in Table 1 and were organised under five major themes I) perceived causes of stillbirth II) prevention practices for fetal loss III) symptoms preceding stillbirth IV) consequences and impacts of stillbirth, and V) treatment seeking and coping mechanisms after stillbirth.

**Table 1. Explanatory model for stillbirth–themes and sub-themes identified among responses from respondents in Kabul province, Afghanistan 2017.**

| Theme | Sub-theme |
|---|---|
| **I. PERCEIVED CAUSES** | • Biomedical<br>• Spiritual and supernatural<br>• Extrinsic factors (actions or behaviours)<br>• Mental wellbeing |
| **II. SYMPTOMS** | • Physical symptoms (pain, bleeding)<br>• Non-physical (knowing or a feeling that something is wrong)<br>• No symptoms |
| **III. PREVENTION PRACTICES** | • Self-care (avoid heavy physical work, eat well, seek care from professional)<br>• Religious rituals<br>• Superstitious practices<br>• Social restrictions |
| **IV. CONSEQUENCES OF STILLBIRTH** | • Psychological impact and grief<br>• Physical health<br>• Social consequences and differential treatment |
| **V. TREATMENT SEEKING AND COPING** | • Actively seeking care for subsequent pregnancy<br>• Consoling with partner or close family<br>• Reassurance of getting pregnant again |

## Theme I—Perceived causes of stillbirth

When asked about the causes of stillbirths, respondents referred to causes which we broadly classified as biomedical, spiritual and supernatural, extrinsic factors, and mental wellbeing (Fig 1). Beliefs of the cause of the stillbirth were not confined to one category, with most respondents mentioning several causes when sharing their beliefs about what they thought could lead to stillbirth and what they have heard as illustrated by this father's comment and in Table 2:

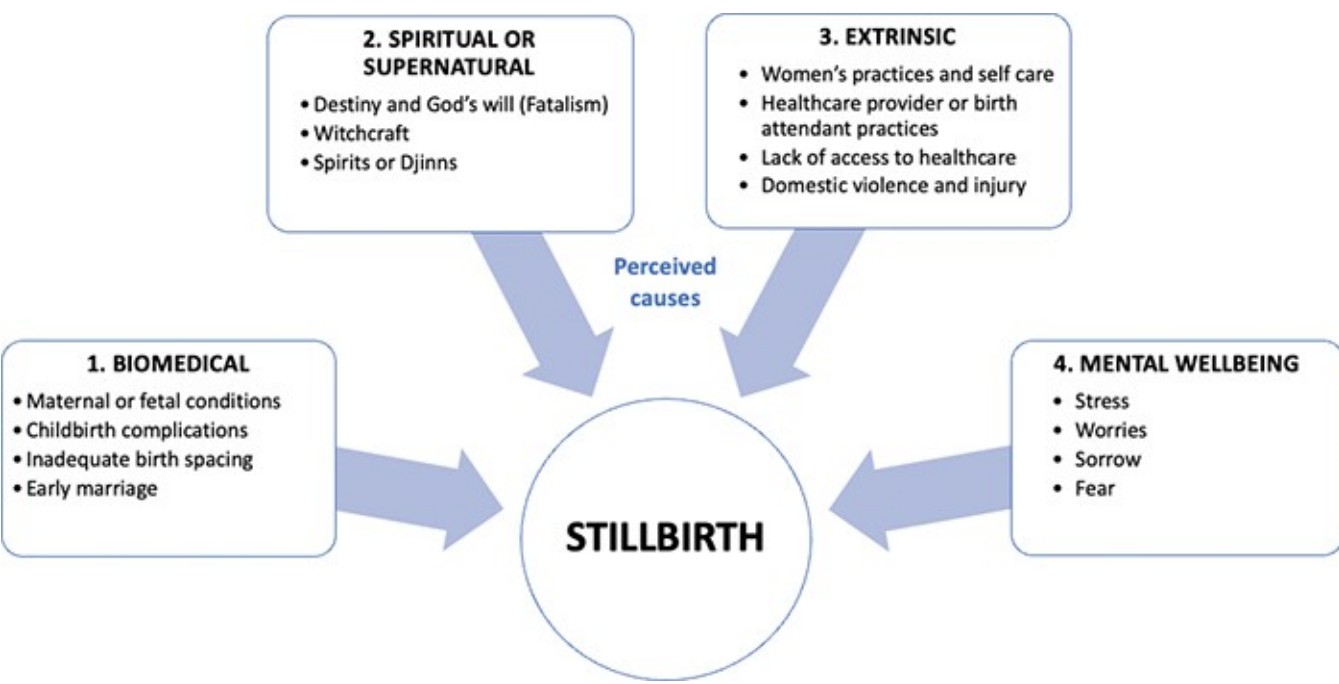

**Fig 1. Theme 1 –Explanatory models of stillbirth in Kabul province, Afghanistan 2017.**

**Table 2. Perceived causes of stillbirth among participants in the study, Kabul province, Afghanistan 2017.**

| Sub-theme | Perceived cause (sub-category) | Examples of perceived cause mentioned | Quote |
|---|---|---|---|
| **Biomedical** | **Maternal or fetal conditions** | Anaemia, "weakness", "lack of blood"<br>High or low blood pressure<br>Reduced amniotic fluid<br>Problem with the baby | *The reason is the blood pressure, anaemia or someone has done black magic over her and sometimes mothers lift heavy weight.*<br>-Mother#11<br>*Some of them say that they had lots of work to do, and it was the cause for the stillbirth or had low blood pressure. But others say that it was due to the anaemia and high blood pressure.*<br>-Mother#18<br>*However, we cannot say anything with the God wishes. . .it was possible my baby was died himself or he might have some health problem.*<br>-Mother#01 |
| | **Problems during childbirth** | Birth asphyxia | *The reasons are such as getting suffocated during the delivery or weakness. Besides, the mother might suffer anaemia, can't circulate the blood well, or her water might have dried which all could cause stillbirth or neonatal birth.*<br>-Female elder#01 |
| | **Early marriage** | Physical maturity of woman | *. . .the early marriage of the girls is another reason of the stillbirth. This is something common that they get married a girl about 15 years old, she doesn't complete her legal age and her womb is not ready to nurture the baby.*<br>-CHW#02 |
| | **Birth spacing** | Consecutive pregnancies deplete mother's nutrition | *The main reasons of these cases are the repeated deliveries once in each year, because the mother faces anaemia and she also feeds her other child with her breasts or from her milk the fetus also feeds from the blood of her mother in her womb. . .*<br>- CHW#02 |
| **Spiritual or Supernatural** | **Fatalism** | God's will<br>Divine destiny | *We also became upset, but it was Allah's pleasure, and we want blessings from Him. It was given to us by Allah and He took it back.*<br>- Father#01<br>*While we did so, as we were able to do, doctors also give us the satisfaction about health of the baby, but it was the willing and destiny of Allah. However, I don't know exactly about the reason it may face something on the way to hospital or doctors were careless about her.*<br>-Father#08<br>*I don't want to tell something about the destiny and willing of God that what He will do, but every woman should try her level best. She should try her level best, later God knows what to do. We accept whatever God wills, because it might be in our destiny.*<br>-Mother #05 |
| | **Witchcraft** | Woman is "bead or thread-affected"<br>Mother receives or affected by an amulet or *taweez* that is usually given to pregnant women by a *Mullah* (religious leader) as protection.<br>Driven by jealousy or revenge | *There are some reasons, for instance: when there is a wedding ceremony, a pregnant mother will lose her pregnancy when she returned back to home because there was a woman at the ceremony who had a bead's string, bead or she already drink the water of the bead, that is why she lose her pregnancy or baby.*<br>-Father#04<br>*. . .people say that she might be affected by the black magic or someone has done talismans (Taweez) on her. But a number of people say she might have had a disease such as me or blood pressure or another disease. People have different beliefs.*<br>- Mother#10<br>*Some people say somethings shadow fell on it, or it might have been affected by someone writing an amulet for it. Some people say it had a sickness like mine, or high blood pressure. The people have different beliefs.*<br>-Mother#15 |
| | **Spirits or Djinns** | Mother affected by spirits (good or evil)<br>Being in proximity with someone who died (or their spirit) or someone who has had a stillbirth | *. . .people said I might have put my foot over a Djinn. . .I didn't believe such things a lot*<br>-Mother#13<br>*It was death shadow to me, my father-in-law died and there was crowd in our home, my father-in-law died, and I was on his head (this means that she was present when he was passed away). I became sick on the third day of my father in law's death ceremony and my child was born dead.*<br>-Mother#06 |

*(Continued)*

**Table 2.** (Continued)

| Sub-theme | Perceived cause (sub-category) | Examples of perceived cause mentioned | Quote |
|---|---|---|---|
| Extrinsic | Women's practices / self-care | Engaging in physically demanding work<br>Not eating well<br>Sudden standing<br>Running | *I think. . .she, is pulling up something heavy or doing hard work at home, that is why she is losing the baby*<br>-Father#04<br>*Personally, may God bless my father! He told me that I have done heavy chores which caused this problem. In fact, I was lifting heavy buckets of water. We didn't have water, so I was going to bring the water. I thought that would be the problem.*<br>- Mother#18<br>*She was doing the heavy chores at home, but I prevent her, but anyhow, she does so. According to the habit of the villagers, women are cooking bread at home in (a dug in ground oven), cleaning the floor, washing the clothes. On that day she washed the clothes, but my mother let her to wash the clothes, and thought that she may never face with any problem, because we have done the same activities during our pregnancy as my mother said. . .Mostly the baby dies during washing the clothes.*<br>-Father#06<br>*A stillbirth is due to many reasons such getting thread affected, sound affected, the mother breaks her diet or lifts heavy weight.*<br>- Female elder#02 |
| | Health provider or birth attendant practices | Inadequate care and attention from birth attendant<br>No proper delivery<br>Prescribing harmful medications | *We are not saying why it happened this way and that way, but there was a little lack of care from the doctors' side.*<br>- Father#03<br>*When I myself was supposed to give birth to my first child, I went to a doctor and she gave me a pill to take. I took it and miscarried my child.*<br>- Female elder#02<br>*There are different causes such as high blood pressure, lifting heavy things, no proper delivery. These are all the causes which make the baby die.*<br>-Mother #21 |
| | Access to care | Not seeing a doctor during pregnancy<br>Giving birth at home | *The reason is the blood pressure, anaemia, sometimes the mother lifts heavy weight. Or the baby is delivered dead due to the lack of doctor's attention, or it is delivered at home*<br>- Mother#13 |
| | Injury or violence | Physical abuse from husband<br>Accidents<br>Earthquake<br>Sudden impact with object | *A lot of people would say she takes after her mother or her sister-in-law as they would also miscarry their children. Family violence is also very much in the families.*<br>-Mother#03 |
| Mental wellbeing | | Stress, worries or sorrow<br>Fear—from earthquake or being frightened by something | *When the battle started in our area, I got pain and delivered the baby.*<br>–Mother#12<br>*It is also either thread affected, or sound affected. The one who has swallowed bead's water has a heavy voice, or she might have been scared at night. One of my brides, may she live long, had taken out the cow to water; when it ran, she was scared and miscarried her twins.*<br>-Female elder#01 |

> . . .*but people are talking about its reasons* (for stillbirth); *she* [mother] *might have lifted something heavy, she might be afraid, or her birth or labour was closed* [referring to black magic].
>
> -Father#09

## Biomedical

We found that most respondents would first refer to biomedical reasons for the cause of stillbirth. Despite also believing in supernatural causes and fatalistic views, there was still an

overarching belief that there were also underlying biological causes and that it was not completely beyond control. Many respondents did not accept the loss without question and searched for reasons to understand why it happened. There was acute awareness of the impact of the woman's health on pregnancy loss and one of the first reasons mentioned by respondents when it came to their perceived causes of stillbirth, was illness in the woman (Table 2). Several respondents spoke about "weakness" or "lack of blood" referring to anaemia, and also high blood pressure and diabetes. As one woman explains,

> *The reason is the sickness of the human itself, her weakness or anaemia that causes the baby to be delivered dead*

> -Mother#14

Early marriage and subsequent pregnancy when young women were not adequately developed was also mentioned as a cause, as was the issue with birth spacing. One father thought that because his stillborn baby had shorter spacing between pregnancies than his previous children that this might have played a part in his child's death. Other biomedical reasons stated were problems during the childbirth including birth asphyxia (Table 2).

## Spiritual or supernatural

The belief that stillbirth was attributed to a higher power was a common view among respondents but also seemed to be a coping mechanism. This sub-theme comprised of three categories–destiny and God's will, witchcraft, and spirits and Djinns (mythological creatures in Islamic mythology).

## Destiny and God's will

There was a sense of fatalism with some respondents attributing the death to "the hand of God"; as one father said, *Allah granted us with a son and took him back from us* [Father#07] or their fate, as this mother stated: *It was definitely in my destiny* [Mother#18]. Most parents were not given a reason for their babies' death by healthcare providers and who often themselves told parents it was God's will. One CHW comment illustrates this view and a sense of acceptance of the death without question as if it was inevitable,

> *People use another proverb: when a tree takes out the flowers, some of them produce the fruit and some others lost—these are the words which are used by the people in the community, then we take the dead baby and came back to home, no one asks the doctor about the reason, why it died during the delivery.*

> CHW#03

## Witchcraft

The second most dominant sub-theme raised as a cause was the use of witchcraft. Women spoke of being "thread" or "bead" affected. This referred to an item–usually an amulet or *taweez*–(thought to hold some magical power and is usually protective but can also be harmful to the person receiving it or being near it). This *taweez* contains written verses from the holy Koran that are either kept in an amulet or written on small pieces of paper and placed in water or a tea pot to drink. Respondents spoke of two types of people that provide these–genuine mullahs (religious leaders) or similarly educated religious persons, that give the woman such

an item to bring protection or a good outcome for her pregnancy, and others (thought to be capable of black magic) considered 'evil' who appear to be providing the same blessing as a genuine mullah but have the opposite intention. A woman may unknowingly receive this item that she wears on her body and ultimately loses her baby as a result. If this occurs, it is said the woman's pregnancy has been "closed" and this can happen to a pregnant woman, or can prevent a woman from conceiving:

> I remember that once she [his wife] *told me that my birth was closed by someone, I don't believe such kind of opinions, but doctors also told me the same words. . .Some of the women go to the magician to close the birth of that woman or do something so the woman should not get pregnant. . .She wasn't able to give birth to the baby and it remained in the womb of mother.*
>
> -Father#09

Sometimes this is done by others as revenge or out of jealousy. This same father relays his wife's belief that the reason for their stillborn baby was a result of witchcraft carried out on the part of his brother's wife:

> Here lives my brother's wife, she is widowed—her husband died four years ago. . . they are living here next to us, and I am supporting them and provide them their daily expenses. So, my brother's wife hopes that my wife will die, and she will get married with me. I don't know, if it is true or false. . .These are the thoughts of another; we should not lose our belief by others' saying.
>
> -Father#09

There was also superstition about women whose pregnancy ended in fetal loss and that she can cause another woman to lose her baby if she is in possession of this "string"; or conversely, if she has lost a baby and is in close proximity to a woman who is pregnant—her pregnancy will be protected while the others will be affected, as this one woman explains,

> . . .*For example, you lost your child then you have a string, if you came to my house for any need, now your string will affect my pregnancy. It means I will lose my child, but you will not.*
>
> - Mother#01

Witchcraft was predominantly perceived to play a role by female elders and CHWs (also quite elderly in this study). Although several respondents did not believe in this kind of magic, they would still consult a Mullah to receive protection. Some parents even spoke of doctors telling them that such magic or witchcraft was responsible, as this father describes:

> She wasn't able to give birth the baby and (it) *remained in the womb of mother, the doctor also said to my wife, your birth is closed by someone and sometimes it also closed unwillingly. But I don't know if it was closed by someone else or closed unwillingly.*
>
> - Father#09

Such black magic is thought to affect the women and her pregnancy only up until the baby begins to move in the womb. For this reason, pregnant women are prevented from going outside the home in this time period.

## Spirits or Djinns

Another frequently held belief was that supernatural powers were at play and being affected by spirts -both good or evil (*Djinns*). Several women spoke of being affected by, stepping on, or passing near a "death shadow"—the spirit of someone who had recently died. This may be an evil spirit; but again, many women, although having heard of these stories, did not believe they were true:

> *It might be a magic or Djinn. I was also told the same—that I was affected by the Djinn. I have heard many things like this. . .I don't believe such things at all, and I am not afraid of such things either.*
>
> -Mother#18

## Extrinsic factors

Several other causes described by participants were grouped as extrinsic factors–being factors that were either intentional or unintentional actions, behaviours or incidents that were perceived to have adverse effects on the pregnancy, fetus or on childbirth. This includes actions of the woman, husband, health provider, including (domestic) violence or injury.

## Women's practices and self-care

One of the most frequently mentioned cause of stillbirth by all respondent types was women engaging in physically demanding work while pregnant. For the most part, this included household work involving "heavy chores" including farm labour, (manual) laundry and cooking, which required lifting or carrying heavy things or bending over.

> *. . .I think I have done heavy work and I might lift a heavy sack which cause its death. I had severe pain and bleeding and then I visited Isteqlal hospital with my husband.*
>
> - Mother#07

Women continued to work either because they had no choice, did not want to upset their in-law's, or were unaware that it may cause harm to their pregnancy. In traditional Afghan households the wife or daughter-in-law is usually responsible for maintaining the household of her in-laws and many women spoke of chores (that were physically demanding) that needed to be done and not wanting to upset their mother-in laws, some knowing that it was bad for their pregnancy. On the other hand, there were some women that believed the pregnancy should not prevent them from carrying out regular activities and did not seem concerned. Men also mentioned that women neglected taking additional care during their pregnancy and continued to carry out heavy chores or work in and outside of the house,

> *The women are careless about their health, my wife is kneading the flour, cooking the bread in the oven, she is going to the grape garden and collecting the grapes, washing the clothes, she is washing the clothes daily, but she is not to blame for that because the children dirty their clothes, and the mother must wash the clothes. . .*
>
> -Father #09

## Healthcare provider practices

Several respondents mentioned problems during the birth or not having a "proper delivery" as a cause, with health providers lack of care and attention contributing to stillbirth. The idea that

there were other outside factors responsible in addition to God's will is demonstrated by one father's comment,

*'Our child was so healthy. . .The doctors were so careless in this regard, but I haven't complained about them. Even if I have complained about them, there was no one to complain to. . .'.*

- Father#02

## Lack of access to healthcare

There was some level of awareness of the importance of seeking care during pregnancy and birth among respondents and that residing near a health facility can be protective while home birth can be a cause of stillbirth (Table 2). This father acknowledges his neglect in this regard to ensuring care for his wife during her pregnancy because of the distance of health facilities, noting from his personal observations that fewer stillbirths occurred in villages where there is good access to care:

*The men are also to blame because we don't bring them* (the women) *to the doctor at the time for the check-up. The doctor might be (far) away from us, or we are lazy to bring them to the doctor, however, those who live in the bazaar* (market) *or near the clinic, they don't have such a problem, because they always go to the clinic and visit the doctors. People of Shikhan village live near to the clinic and they don't have a case of the stillbirth.*

Father#09

## Domestic violence and injury

Partner and family violence were alluded to by several respondents as being very common within families. Female elders and CHWs raised this issue frequently, however, some women also mentioned that family violence can contribute to pregnancy loss. This was also believed to affect women's emotional state, which in turn was perceived as leading to stillbirth or miscarriage.

*. . .most of those whose children are miscarried are said to have undergone family violence. Somebody fights or gets scared, or their husbands behave badly with them and so on.*

Mother#03

## Mental wellbeing

A woman's mental wellbeing or emotional state was also thought to play an important role in pregnancy loss among respondents in our study. Most respondents spoke of the effect of stress, worries, sorrow and fear or shock on pregnancy loss. It was believed that if a woman was frightened or scared it could cause her to have bleeding and lead to stillbirth or miscarriage.

*My first baby was six months and it died in my womb. I don't know why it has died, but I have been afraid of the cat.*

- Mother#08

The ongoing conflict from the war also contributed to high levels of fear and stress that was thought to also cause pregnancy loss as one CHW explains:

*God knows that my daughter-in-law was affected due to the thread (witchcraft) or fear of the rocket* [from the ongoing conflict]. . .

- CHW#01

## Theme II—Symptoms

When women were asked to describe the problems they encountered either during pregnancy or childbirth and when they knew or realised their baby had died, we found that women either experienced physical symptoms, which signified that there was a problem, or they stated that they had a feeling and felt that the baby had died–most often due to a perceived absence or cessation of any movement. There were also women who were not aware of any problem until after the birth when they found out their baby had died. Many reported experiencing physical (body) pain or bleeding prior to losing their baby as these two women explained,

*I didn't know. I was cleaning the house and I suddenly got a body ache, but I thought that was because I had lifted heavy things.*

-Mother#09

Several women were aware of their baby's movement and perceived any reduced movement as indicating a problem or that their baby had died,

*Yes, it* [the baby] *was moving and I knew it was alive. When I touch my womb, I knew it was alive. I knew it too when it was lost* [died].

Mother#02

*They* [the doctors] *said that it was a baby girl and was healthy, but when I returned home, I told my mother-in-law that it wasn't alive because I didn't feel any of its movements and it turned around as I was sleeping.*

–Mother#08

## Theme III—Prevention practices

Respondents reported a variety of practices during pregnancy to protect themselves and their baby from harm or adverse outcomes. Most of these practices were influenced by beliefs around the perceived causes of stillbirth or miscarriage and avoiding these if possible or taking action to avoid them. This comprised i) self-care practices including not engaging in heavy physical work, eating well and receiving vaccinations; ii) religious practices such as visiting a Mullah–for both prevention and treatment to avoid being affected by black magic, and iii) social restrictions such as avoiding going out a night and avoiding men, due to prevailing superstitious beliefs around witchcraft (Table 3).

### Self-care practices

Despite prevailing fatalistic beliefs, women felt they had some control in ensuring a healthy pregnancy and that there were actions they could take, as explained by one woman:

**Table 3. Prevention practices to avoid stillbirth reported by study participants.**

| Sub-theme | Practices | Quote |
|---|---|---|
| **Self-care practices** | Avoid heavy physical work<br>Eat well<br>Visit healthcare provider in pregnancy | *She should not lift heavy things and have good food. She should have more fruit and make a schedule for her meal in order to have everything every day not one thing every day.*<br>*-Female elder#01* |
| **Religious rituals** | Visit the mullah/religious leader<br>(for prevention and treatment) | *But as I went to Mullah. . . he said that I definitely have Djinn on my back and I was told that after your 40 days of pregnancy, bring a sheep for me and I will make an amulet for you from the blood of its liver so that your babies survive. I was told such things.*<br>*-Mother#08*<br>*If a pregnant mother finds bleeding after the sixth month, she goes to the Mullah, then he spell-binds the lock to keep the pregnancy safe and prevent her from losing it (the baby). When she gets to the ninth month, she goes to the Mullah to open the lock and then she has a normal delivery. I have heard these things, and this is true.*<br>*-Father#09* |
| **Superstitious practices** | Wear a protective amulet<br>Avoid entities associated with bad luck or black magic | *This string is used in the magic bead, and women keep it to themselves for their own protection, this is something common in our community. . . This is a harmful bead, and this string is putting inside the bead then women keep it for themselves from the protection of other women's strings. My sister has gone to the mosque for the religious ceremony on the 10th of Moharam, there were women from Pashayee village as well and most of them have strings with themselves, so those strings affected the pregnancy of my sister and she miscarried her three months pregnancy. . .*<br>*-Father*<br>*. . .She (mothers) should not go under a tree at night, because the trees have black magic and there are many djinns that affect the baby and the mother. Sometimes people say the mother doesn't have luck.*<br>*-Mother#03* |
| **Social restrictions** | Not going outside home especially at night<br>Avoid younger women<br>Avoid women with a recent pregnancy loss<br>Avoid sexual intercourse<br>Avoid being near or in contact with other men | *During the pregnancy the mother should be apart from her husband in order that the pressure is not happened over the baby and it should not be miscarried.*<br>*-CHW#02* |

*She* [pregnant women] *should receive vaccination and visit the clinic, doctor or a midwife. Why should her baby be miscarried if she takes care of herself? I don't want to say something about the destiny and willing of God and what He will do, but every woman should try her level best. . .later, God knows what to do. We accept whatever God wills, because it might be in our destiny.*

-Mother#05

Whereas some respondents were aware of the importance of good nutrition, avoiding demanding physical work, and consulting with a health provider, others did not know exactly what sort of food was needed in pregnancy,

*Actually, I don't know what a pregnant woman should eat, but I know a little that she shouldn't do heavy chores and lift heavy things. She should eat fruits and shouldn't go outside the room alone during the night.*

-Mother#09

## Religious rituals and superstitious practices

Religious rituals and the use of *taweez* were common during pregnancy to keep the baby safe or as a preventive measure if women feared they had been "cursed" or had been in the vicinity of someone with the ability to do this, and also when complications occurred. Women would visit a Mullah on any of these occasions. One father explained how consulting a religious leader is done to ensure a healthy pregnancy outcome, with the positive outcome reinforcing the practice,

*I haven't gone* [to visit a Mullah], *but my cousin has gone to a Mullah, and he received an amulet and the Mullah told him that your wife's delivery has been closed by someone; he received an amulet, and his wife had a normal delivery.*

Father#08

And that if seeing a doctor was not possible, then at a minimum, woman should consult a Mullah,

*. . .if you are not able to go to the doctor, you should go to the Mullah and receive an amulet.*

Father#08

Oftentimes the symptoms or signs of complications are perceived to be the result of causes other than biomedical reasons which leads to delays in care-seeking. For instance, when a pregnant woman presents with symptoms of high blood pressure such as seizures, some respondents believed that this was due to the woman being possessed by spirits. Healthcare providers also reported the community beliefs of women being haunted, as these midwives explain:

*Most of the children who die is because the mother has palpitations or faints. The community people say she is haunted and take her to a Mullah. She is palpitating when she is taken to Mullah and her treatment is delayed. So, her time is wasted. In the suburbs* (rural areas), *especially, the people say she was haunted and scared. That is why the child died. I don't agree with this custom and belief of the people because the mother is suffering high-blood-pressure and they mistake it as being haunted.*

-Midwife#04

*. . .when the baby is dying because of the blood pressure of its mother, the people thought that it's because of the Djinn* (bad spirit). *In reality, the mother has experienced a complication.*

-Midwife#01

## Social restrictions

Respondents explained that pregnant women should never be alone, especially at night, and in particular, avoid standing under a tree–again, for fear of witchcraft. It was also stated that young girls in particular should not visit pregnant women, and pregnant women should avoid visiting or being visited by other women who have had a stillbirth or pregnancy loss within 40 days of that loss, due to the fear that it might affect that women's pregnancy or newborn. Respondents spoke of this potentially affecting the growth of baby and could lead to deformities,

One CHW from a rural area explains about a woman who has had a recent stillbirth or pregnancy loss:

"No, they never let the mother enter (the house), even they fight with her (and say) why did you come here before you completed the "40 days shadow" of your baby's death? Or they immediately give her bread with salt or bread with sugar to eat, because her being will (then) not affect their newborn baby."

CHW#03

Generally, men were not permitted near women who were pregnant and any contact with males, including husbands, was avoided including sexual intercourse during pregnancy.

## Theme IV—Consequences of stillbirth

The main impacts of stillbirth described by participants concerned the psychological effects and grief it caused for parents and the family in general and the various ways this manifested, the physical effect on women's health, and the social implications for women and how they are treated by their family or communities.

### Psychological impact and grief

The psychological impact of stillbirth and grief on both women and men was clearly evident in parents' descriptions of how they felt and what they experienced after the stillbirth. Grief manifested in many forms including sadness and longing, anger, disbelief, and disappointment. Parents spoke of feeling sorrow and longing for their child, as this father reflected,

*I wished it were alive and lived with us at home. My heart wouldn't hurt because of what happened. . .*

-Father #03

Another father indicated how it affected him and his wife,

*Of course, it had a bad effect on the parents, it creates mental pressure because the baby has gone, and we are also concerned about the mother. I feel concern while I'm thinking about my son; we were waiting for nine months and bought clothes for him and selected the name. While these things come to mind it makes me sad.*

Father#08

Anger was also a frequently mentioned emotion by parents:

*I got so upset and angry. One night I wake up to breastfeed the baby, but when I realized that there is nothing, I got so upset and cried a lot, but later I repented to God. Her* [the baby's] *father cried as much as he visited her grave. He is working in a university. He was not at home* (when it happened), *but when he returned home, he immediately visited her grave.*

Mother#05

The intensity of the grief felt by parents is demonstrated in these respondents' comments,

*. . .Now, I always cry for my baby, and I see it in my dreams every night that he/she is talking with me.*

Mother#12

Generally, a stillbirth brings much sorrow for the parents, but some expressed that the wider family and even relatives and neighbours are affected. Parents spoke of their whole families being sad and upset about the loss, as one female elder stated, "*The relatives and neighbours get upset because of a stillborn boy or girl that somebody gets. . .*".

Respondents also spoke of the continuing absence of the child as time passed, as this mother explained,

*As it is God's work, we can't do anything. You can't do anything. You just think if it lived, I would have many children, or he/she would be as big as his/her* [somebody else's child who was born at the same time] *child. . .*

Mother#03

## Social consequences

Although both women and men remarked that pregnant women do not look after themselves adequately or should be taking more care, there did not appear to be severe blame directed at women.

Overall, most respondents–including women themselves—spoke of surrounding family members and communities being sympathetic towards parents who have lost their baby, as one CHW explained,

*People show their sympathy with the parents and encourage them and tell them, this one has gone, but you are young, and Allah may grant you with another baby; here in Afghanistan we have only sympathy and can do nothing with the parents.*

CHW#02

This was also confirmed by women's accounts,

*No, they* [her in-laws] *don't say anything. My mother-in-law is a nice woman and my father-in-law hasn't said anything. It is Allah' willing and we accept whatever is in our destiny.*

-Mother#02

However, CHWs, many who were also elders in their communities, spoke of women being blamed especially among less educated families,

*If they* (their husbands) *were educated, they encourage their wives and show their sympathy and say all these happens by the order of Almighty Allah, thanks to Allah, that you are feeling well and you're alive, please eat energetic food to become healthy. If the husband is a farmer or an illiterate person, he orders his wife to stand up and do your daily chorus, there is neither sickness nor sleeping. It is related to the family, if they are educated people, they* [the women who have lost their baby] *are treated very well, but the illiterate family behaves very badly with their wives.*

–CHW#03

Additionally, there were some differences in how some women were treated in the post-partum period if they had a stillbirth compared with those whose baby had survived. Generally, after birth women rest for a 40-day period, and are given nutritious food to assist with their recovery, as one woman explained,

*I was so ill and weak* (after the stillbirth). *I couldn't do anything, and I felt bad, but my husband and family consoled me and told me it's ok. They insisted I not cry, and they really supported me. I was not allowed to do chores and they were always bringing me medicines and fruits.*

Mother#12

A female elder who visited women after birth also reported the variation in how mothers are treated when they return home,

*. . .I was told to visit the mothers until six days. When I asked her whether her husband brings her medicines, some replied that they are provided with medicines, meat, eggs, oil (dairy, ghee) and other things. But when I visit others, I saw their mouths were dry and they say their mother-in-laws don't allow them.*

-Female elder#01

A CHW also confirmed this,

*No, when you lose your baby, there is neither 20 days nor 40 days. Everything has been finished. . .she must start her daily house chores, because she doesn't have live birth, some of the families cook and provide the special food for the mothers, but some others didn't. If she had a live birth, she must rest up to 40 days, and be provided additional meal like soup, Shira, lity, raisin and Chawa* [a healing soup made with molasses and ginger].

-CHW#03

Some women who did have the option to rest chose not to and went back to their daily routine,

*I haven't rested more than 10–15 days. . . There were chores to do, so I stood up. . .Yes, she* [mother-in-law] *refused me working at home and I was told I might be sick due to it. She added that now we don't know, but later we might have severe pain and you now have the strength, but later you might have pain. But I was still doing some little chores.*

Mother#17

## Theme V—Treatment seeking and coping mechanisms

### Actively seeking care for subsequent pregnancies following stillbirth

Most cases of stillbirth occurred in health facilities as many women were seeking care because of a complication. When we explored with parents what they did after a stillbirth and whether they took any action to prevent a recurrence or identify what may have been an underlying issue, we found most families were more proactive in their subsequent pregnancy. Based on their perceived causes, parents felt there may be some actions or behaviours they could adopt to ensure a healthy pregnancy. As one father stated,

*She* [his wife] *was concerned about why did I lose my baby? That might be the reason—her low blood pressure; I have told her that, Allah is gracious, he may grant us with other children. After that we couldn't tolerate that situation, and we go to that doctor who gave her injections and medicines and with the bless of Allah this daughter lived and we did not lose her.*

-Father#04

As a consequence of the first experience, most women reported taking additional care during their next pregnancy,

*I was so angry on them* [the babies she lost] *after their death and I was doing heavy chores, but God blessed with this baby. After that when I got pregnant (again), I had an apple, a glass of milk and other things and visit the doctor. I visited the clinic for vaccination. I was telling my problems in order to have a healthy baby in my womb. I was taking much care of myself. I was visiting the midwives when I got pregnant, and I was visiting everywhere.*

Mother#05

Women also expressed that they would like to know what caused their pregnancy loss so they could receive treatment as this woman stated,

*It would be better if I know the cause for it. If I do, I will treat myself.*

-Mother#17

## Consoling with partner or family

In regard to how parents coped afterwards with the stillbirth, most parents in our study spoke of sharing their grief and consoling one another and sometimes with other family members and through reassurance that they will get pregnant again. Parents tended to be supported in their communities including relatives and neighbours who showed sympathy for the parent's loss. Women did not discuss this with other men apart from their husbands, but shared their sorrows with other women,

*I shared my concerns with my husband and all the members of the family. Later my husband and members of the family told me not to worry, because I have a son and Allah will bless you with another son. But I had no patience. . .*

-Mother#12

Men, on the other hand did not share grief with other men and only spoke with their wife,

*First, he (her husband) shares it with me, and I share it with myself or him. My husband doesn't have good relations with my in-laws to share his grief with them.*

-Mother#09

*No, men do not talk to other (men), he talks with me as I am his wife. For example, when he sits, he tells the story, he regretfully was saying "if our child had not died what would have happened. . ."*

-Mother#01

### Reassurance of getting pregnant again

Most parents consoled themselves with the idea they can have more children, and this was also something communicated by surrounding family members,

> . . .My husband was upset too, but not as much as me. He expressed his sympathy and said that Allah will bless us with other children.
>
> -Mother#02

## Discussion

Our findings provide insight into the perceived causes of and explanatory models for stillbirth that exist in Afghanistan. The results indicate that individuals ascribe multiple causes to stillbirths, from biomedical to supernatural factors, as well as extrinsic physical or behaviours, and mental wellbeing. Prevention practices in pregnancy aligned with perceived causes and included engaging in self-care, religious rituals, superstitious practices and adhering to social restrictions. Symptoms preceding the stillbirth reported by women included both physical and non-physical symptoms or none at all. The impacts of stillbirth revolved around the psychological effects and grief, the physical effect on women's health, and the social implications for women and how they are perceived by communities.

Perceived causes of stillbirth have not been explored explicitly in many low-income settings, however, there are similarities in our findings to what has been reported in studies that have examined experiences of pregnancy loss. In rural eastern Uganda, women, grandparents and traditional birth attendants also referred to biomedical and spiritual causes, as well as physical causes of stillbirth including domestic violence [17]. A study in rural Ethiopia found most women believed malevolent spirits were responsible for stillbirths and early neonatal deaths [18] while in Nepal, God's will was a frequently mentioned factor contributing to perinatal loss with respondents referring to a loss being a result of the "wrath of God" [19]. In rural Kenya, adverse pregnancy outcomes were also attributed to not following certain behaviours during pregnancy and, similar to our study, spirits of the deceased were believed to also lead to pregnancy loss [20]. Fatalistic beliefs and attributing deaths to God's will was referred to by respondents in our study, but not in relation to the woman having done something against God. Comparable explanations were given to miscarriage among Qatari women who also identified supernatural forces, medical conditions, as well women's emotional state as key causes of miscarriage [21]. Additionally, in Kenya, perceived causes of miscarriage included both biomedical reasons or punishment for not conforming to cultural norms [22].

Importantly, our findings highlight that many respondents believed stillbirth was preventable, which is encouraging for the effectiveness of future community education. Further, many women expressed that they had not received any information on how they can ensure a health pregnancy and were interested in receiving this. Currently in Afghanistan there are no stillbirth prevention messages delivered to women or their families during or prior to pregnancy. Such messages could be incorporated into the training curriculum of CHW who conduct household visits to pregnant women in Afghanistan and also into antenatal care training packages for health care providers to ensure these messages are given during antenatal care visits. It would also be critical to disseminate such messages to the broader community to address prevailing misperceptions and influence key decision makers in the household [23]. Religious leaders can be engaged in delivering education given the reliance on them in the community for advice and treatment when facing problems during pregnancy. Other means of conveying information about stillbirth to other family members including husbands and mothers in law

that are not restricted only to pregnant women and using multiple forms of communication should also be explored.

Many respondents in our study appeared to be aware that a change in their baby's movements signified a problem, however, they did not always act on this. It would be important to include messages in health education efforts to encourage women to seek care as soon as they felt a change, as reduced fetal movement is associated with stillbirth [24]. These could be during antenatal care visits or through larger community-based educational campaigns similar to those that have being developed in high income countries [25, 26] which can be adapted for low-income settings such as Afghanistan. Some key recommendations for policy makers to consider for implementation are outlined in Box 1.

## Box 1. Recommendations for increasing awareness and disseminating messages around stillbirth

**Key Recommendations**

**1. Deliver stillbirth prevention messages at the community level**

• Incorporate key messages that support stillbirth prevention into antenatal care counselling packages to be delivered by health workers adapting from existing evidence-based packages [26]. Key messages may include the following:

○ Stillbirth is preventable. Good quality antenatal care and giving birth in a health facility can reduce stillbirth risk by checking the baby's growth and identifying the best timing of the birth. Stillbirth risk cannot be increased by having contact with other women who have had a pregnancy loss.

○ Seek care immediately if there is any change in the normal pattern of the baby's movement especially from 28 weeks of pregnancy.

○ Don't delay seeking formal healthcare if there is any concern that anything doesn't feel right.

○ Attend antenatal care as early as possible in the pregnancy to identify and manage any problems early.

• Ensure healthcare providers discuss the possibility of stillbirth with women, including important danger signs and what to and where to go if they notice these.

• Consider developing community level campaigns to educate women and the wider community including men and parents' in-law on stillbirth prevention to dispel myths around these deaths.

• Engage and sensitise religious leaders about stillbirth and its prevention to ensure they communicate accurate information to women and families

**2. Ensure healthcare providers are trained adequately on stillbirth**

• Include stillbirth prevention and management in training curriculum for doctors, midwives, nurses and community-level workers. Education should include information on the preventability of stillbirths to dispel myths and incorrect beliefs surrounding these deaths.

> • Ensure training curriculum includes adequate content and information on stillbirth including the care of women after a stillbirth or newborn death. Available toolkits and resources can be adapted [27, 28].
>
> • Identify what kind of support women and families who lose a baby as a result of a stillbirth or neonatal death need in Afghanistan including psychosocial and bereavement care that can be incorporated into post-natal care.

The finding that some health care providers reportedly told family members that the loss of their baby was also due to witchcraft or other superstitious practices was concerning. This indicates that health care providers also need to be targeted for health education messages around stillbirth prevention and further research into their knowledge and perceptions is also required. Healthcare providers' views and the information they provide to parents about stillbirth either prior to or after childbirth has not been comprehensively explored in the literature, particularly in low-resource settings. A survey of healthcare providers in Qatar and the UK that assessed knowledge and attitudes of healthcare providers found that only 18% believed stillbirth was preventable and in Qatar this was 41% [29]. Another global internet survey also found prevailing belief among health professionals that stillbirth was not preventable [30]. These and our findings suggest that greater efforts are needed to raise awareness of the stillbirth causes and prevention practices to ensure that healthcare providers are delivering the correct information. In some countries including Australia, assessment of the midwifery curriculum for stillbirth education identified inadequate time and information and variation across universities on what is taught about stillbirth and that standardisation is required [31]. Such a review would also be beneficial for Afghanistan.

In terms of prevention practices in pregnancy our findings were comparable to the literature including the use of protective witchcraft and preventive practices revolved around religious performing rituals [19]. Restrictions on social mobility and avoiding men during pregnancy is commonplace in many countries including in Kenya as was avoiding heavy physical activity and following prescribed dietary practices [20, 32]. The main concern with some of these practices are its contribution to delaying care seeking from formal care providers and attendance at antenatal care visits, for example. It would be important to consider how to develop messages to address this without dismissing cultural or social practices that were valued or important for community.

The coping mechanisms of families in our study were similar to what is documented in studies in similar settings with support obtained predominantly from spouses and family members as there is a general absence of formal support services available in low-resource settings. Reassurance of future pregnancy and focusing on the next pregnancy was a frequently reported coping method and also reported in an Indian study [33]. It was not clear in our study or in others what support services or structures would be helpful or beneficial for women and their families after a pregnancy loss and this needs further research.

Although women who experienced a stillborn baby reported that their families and communities were sympathetic, there appeared to be some level of stigma towards women who had given birth to a stillborn as those who were newly pregnant or trying to conceive were advised to avoid such women. The literature documents the myriad of ways in which women who have had a stillbirth are stigmatised in various societies and contexts, ranging from being rejected, divorced, and the impact it had on the value of women as mothers [17, 18, 34, 35].

Although these extreme forms were not prevalent among our respondents, efforts are still needed to dispel misinformation to reduce any stigmatisation of women who have a pregnancy loss in Afghanistan. Additionally, in our study women who had lost their baby did not receive the same treatment or period of rest and care as women who had a live born baby suggesting that their experience of childbirth and their identity as a mother is not recognised to the same extent, despite undergoing full physical childbirth and would therefore require the same recovery period. Other studies have shown how women who experience a pregnancy loss often feel this loss of identity and absence of recognition of their motherhood [35].

A key limitation of our study is that data collection was confined to only one province in Afghanistan which is the most progressive and urbanised and most of our respondents had given birth at a health facility. We were also limited to rural districts that had good access to health facilities and medical care all of which may limit the generalisability of our findings. As interviews were conducted after the experience of stillbirth had occurred, respondents' beliefs about causes and future preventive behaviours may have been influenced by what they were told by their healthcare provider and should be considered when interpreting the findings. In addition, our sample of female elders was relatively small, and we may not have adequately captured the range of existing beliefs. Further research is needed to understand experiences from different parts of the country particularly in very rural areas where access to medical care is much lower and among a range of different ethnic groups.

## Conclusion

Our findings are the first to understand the etiological explanations of stillbirth from a local Afghan perspective and suggest that these perceptions are important to consider when developing health education messages for stillbirth prevention. Health education and health promotion for stillbirth prevention in Afghanistan is urgently needed and should target both community and health service providers to support stillbirth risk reduction, dispel misinformation and reduce any social stigma around causes of pregnancy loss and towards women that have a stillbirth.

## Acknowledgments

We sincerely thank all study participants who gave their time to share their personal experiences and insights. Many thanks to our interviewers, Mr Rohullah Sahibzada, Ms Nasreen Quaraishi, and Ms Friba Nasiri for their role in data collection. We are also grateful to the staff at participating health facilities that assisted in various ways during data collection. We would like to acknowledge the Ministry of Public Health and hospital managers for permitting access to the health facilities and Management Sciences for Health, Afghanistan, and their staff for supporting and facilitating data collection in Kabul.

## Author Contributions

**Conceptualization:** Aliki Christou, Neeloy Ashraful Alam.

**Data curation:** Aliki Christou.

**Formal analysis:** Aliki Christou, Camille Raynes-Greenow, Adela Mubasher, Neeloy Ashraful Alam.

**Funding acquisition:** Aliki Christou.

**Investigation:** Aliki Christou, Adela Mubasher, Sayed Murtaza Sadat Hofiani, Mohammad Hafiz Rasooly.

**Methodology:** Aliki Christou.

**Project administration:** Aliki Christou, Sayed Murtaza Sadat Hofiani, Mohammad Hafiz Rasooly, Mohammad Khakerah Rashidi.

**Supervision:** Aliki Christou, Camille Raynes-Greenow, Adela Mubasher, Sayed Murtaza Sadat Hofiani, Mohammad Hafiz Rasooly, Mohammad Khakerah Rashidi, Neeloy Ashraful Alam.

**Writing – original draft:** Aliki Christou.

**Writing – review & editing:** Aliki Christou, Camille Raynes-Greenow, Adela Mubasher, Sayed Murtaza Sadat Hofiani, Mohammad Hafiz Rasooly, Mohammad Khakerah Rashidi, Neeloy Ashraful Alam.

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
