## [Decision Letter · Decision Letter 0]

8 Feb 2023

PGPH-D-22-01805

Explanatory models of stillbirth among bereaved parents in Afghanistan: implications for stillbirth prevention

Dear Dr.Christou,

Thank you for submitting your manuscript to PLOS Global Public Health. After careful consideration, we feel that it has merit but does not fully meet PLOS Global Public Health’s publication criteria as it currently stands. Therefore, we invite you to submit a revised version of the manuscript that addresses the points raised during the review process.

We look forward to receiving your revised manuscript.

Kind regards,

Dickson Abanimi Amugsi, PhD

Academic Editor

Journal Requirements:

1. Please send a completed 'Competing Interests' statement, including any COIs declared by your co-authors. If you have no competing interests to declare, please state "The authors have declared that no competing interests exist". Otherwise please declare all competing interests beginning with the statement "I have read the journal's policy and the authors of this manuscript have the following competing interests:"

3. Please provide separate figure files in .tif or .eps format only and remove any figures embedded in your manuscript file. Please also ensure that all files are under our size limit of 10MB.

4. In the online submission form, you indicated that "The data can be requested from the authors. We did not obtain consent from participants to place the data in a public repository". All PLOS journals now require all data underlying the findings described in their manuscript to be freely available to other researchers, either 1. In a public repository, 2. Within the manuscript itself, or 3. Uploaded as supplementary information.

Additional Editor Comments (if provided):

Reviewers' comments:

Reviewer's Responses to Questions

**Comments to the Author**

1. Does this manuscript meet PLOS Global Public Health’s publication criteria? Is the manuscript technically sound, and do the data support the conclusions? The manuscript must describe methodologically and ethically rigorous research with conclusions that are appropriately drawn based on the data presented.

Reviewer #1: Partly

Reviewer #2: Yes

2. Has the statistical analysis been performed appropriately and rigorously?

Reviewer #1: N/A

Reviewer #2: Yes

3. Have the authors made all data underlying the findings in their manuscript fully available (please refer to the Data Availability Statement at the start of the manuscript PDF file)?

Reviewer #1: No

Reviewer #2: No

4. Is the manuscript presented in an intelligible fashion and written in standard English?

Reviewer #1: No

Reviewer #2: Yes

5. Review Comments to the Author

Reviewer #1: Reviewers comment

This study is really intriguing and may help the people of Afghanistan prevent stillbirths. But I have some reservations about the methodology component that the Authors required to address

1. The authors mentioned that in line 112’ we used a qualitative study design employing semi-structured, in-depth interviews to elicit the…Is that supposed to be an approach? Can you describe the kind of design that was used for this particular study? , we use pronouns such as “I” and “we”. This is acceptable when writing personal information, or a book. However, it is not common in academic writing thus Avoid ‘we ‘ from your manuscript

2. It is unclear in this manuscript how the sample size was adjusted by the authors. Can you explain line 127 and why you only take 21 women? Please provide more information about how the sample size for the in-depth interview was determined.

3. "Interviews were done by three experienced Afghan qualitative interviewers and a foreign female public health researcher," it was written at line 135. Does engaging a foreign researcher improve the quality of the data? Did they comprehend the culture? Could you please explain this?

4. How do you assure the trust worthiness of the data??? Can you include this?

5. One of the main rigor standards used within the qualitative research paradigm is triangulation. Why did the authors choose to conduct this study using just one method of data collection? Could you give your response to this?

6. "We employed the core domains of Kleinman's sickness explanatory model," was stated at line 155. Can you clarify this further by including it in the introduction section?.

7. This paper need extensive linguistic editing.

Reviewer #2: Thank you for the opportunity to review this paper. The authors have conducted a qualitative analysis of data from respondents in Afghanistan in 2017 on perspectives, attitudes and practise related to stillbirth. Seen as a continuum of the author’s work, where they have published the key findings from this work in another paper, this research adds to community beliefs and perceptions on this topic and represents an important contribution to the literature on maternal health.

The paper is methodologically sound and clearly communicates the key messages in its current format. However, issues in places are highlighted, and I encourage the authors to review them.

1. The paper demonstrates encouraging results from the community members who mostly explain stillbirth through a biomedical lens; moreover, despite the country having seen considerable conflict-related challenges that have affected the status of women in society, most responses demonstrated relatively progressive and supportive families where women are by and large taken care of. However, the paper does identify its limitation, but most of the respondents are urban residents who have given birth in health facilities which does colour in most of their findings. Kindly provide this facet earlier in the paper in the data collection stage as the readers required better context to the results

2. Another related aspect that affects the selection of the respondents is whether they were directly chosen from the community or through health system records. As the authors indicated that most respondents came from health system interaction, please clarify why the sampling did not or could not also collect data from those women who had home-based deliveries or who had chosen not to go to health facilities. It is expected that there will be a difference in beliefs and practises between groups of women who had stillbirths at an institution compared to those that did not.

3. Please review table two and the theme of extrinsic factors; some of these can fit under biomedical explanations.

4. Could the authors please explain how the five themes present as results in lines 175 match with Kleinmans’ four categories that the authors used to frame the analysis(lines 159-161)

5. The authors have already shared that they have published another paper from this data set which they reference(13). However, it was not immediately clear to me how this body of work is distinct from their previous publication. Could the authors kindly clarify how this paper is distinct from the previous research findings

6. As part of the presentation of results, Table 3 could be reduced together, removed

7. One of the major implications of their research which I agree with the authors is that healthcare workers holding superstitious views about stillbirth and communicating it to patients is alarming. Kindly consider adding this view to the table on the recommendations in the conclusion discussion section.

6. PLOS authors have the option to publish the peer review history of their article (what does this mean?). If published, this will include your full peer review and any attached files.

**Do you want your identity to be public for this peer review?** For information about this choice, including consent withdrawal, please see our Privacy Policy.

Reviewer #1: **Yes: **Trhas Tadesse Berhe

Reviewer #2: **Yes: **DR Danish Ahmad,MBBS,MSc,PhD,MNAMS,IP-FPH

---

## [Decision Letter · Decision Letter 1]

19 May 2023

Explanatory models of stillbirth among bereaved parents in Afghanistan: implications for stillbirth prevention

PGPH-D-22-01805R1

Dear Christou,

We are pleased to inform you that your manuscript 'Explanatory models of stillbirth among bereaved parents in Afghanistan: implications for stillbirth prevention' has been provisionally accepted for publication in PLOS Global Public Health.

Best regards,

Dickson Abanimi Amugsi, PhD

Academic Editor

Reviewer Comments (if any, and for reference):

Reviewer's Responses to Questions

**Comments to the Author**

1. If the authors have adequately addressed your comments raised in a previous round of review and you feel that this manuscript is now acceptable for publication, you may indicate that here to bypass the “Comments to the Author” section, enter your conflict of interest statement in the “Confidential to Editor” section, and submit your "Accept" recommendation.

Reviewer #2: All comments have been addressed

2. Does this manuscript meet PLOS Global Public Health’s publication criteria? Is the manuscript technically sound, and do the data support the conclusions? The manuscript must describe methodologically and ethically rigorous research with conclusions that are appropriately drawn based on the data presented.

Reviewer #2: Partly

3. Has the statistical analysis been performed appropriately and rigorously?

Reviewer #2: Yes

4. Have the authors made all data underlying the findings in their manuscript fully available (please refer to the Data Availability Statement at the start of the manuscript PDF file)?

Reviewer #2: No

5. Is the manuscript presented in an intelligible fashion and written in standard English?

Reviewer #2: Yes

6. Review Comments to the Author

Reviewer #2: I am recommending the paper be accepted. Editors' overall discretion is warranted as the authors have left some of the reviewers one point for editor discretion

As the second reviewer, while I suggest acceptance for publication,I wasn't entirely satisfied on two points I highlighted here.

First, the paper is rather long, with many quotations and tables provided; while these still add important findings, they affect readability somewhat. However, this point can be overlooked as long as the journal article formatting requirements are met.

The second point is that one of my comments(point #5) to the reviewer was how this paper is sufficiently different from another paper the authors wrote using the same data. While the authors provide a response that I somewhat accept, I would request that the editors view the author's response.The paper doesn't require another round of revision but the two points can be addressed by another view to concur with the responses

7. PLOS authors have the option to publish the peer review history of their article (what does this mean?). If published, this will include your full peer review and any attached files.

**Do you want your identity to be public for this peer review?** For information about this choice, including consent withdrawal, please see our Privacy Policy.

Reviewer #2: **Yes: **DR Danish Ahmad (MBBS,MSc,PhD,MNAMS,IP-FPH)
